# Developmental Cascades Link Maternal–Newborn Skin-to-Skin Contact with Young Adults’ Psychological Symptoms, Oxytocin, and Immunity; Charting Mechanisms of Developmental Continuity from Birth to Adulthood

**DOI:** 10.3390/biology12060847

**Published:** 2023-06-13

**Authors:** Adi Ulmer-Yaniv, Karen Yirmiya, Itai Peleg, Orna Zagoory-Sharon, Ruth Feldman

**Affiliations:** Center for Developmental Social Neuroscience, Reichman University, Herzliya 4610101, Israel; adi.ulmeryaniv@post.runi.ac.il (A.U.-Y.); karen.yirmiya@post.runi.ac.il (K.Y.); itai.peleg@post.runi.ac.il (I.P.); orna.zagoory@runi.ac.il (O.Z.-S.)

**Keywords:** preterm, mother–child synchrony, kangaroo care, anxiety, depression, oxytocin, secretory IgA

## Abstract

**Simple Summary:**

Premature birth disrupts the continuity of bodily contact between mother and newborn, leading to greater prevalence of physiological, cognitive, and behavioral difficulties among prematurely born children. We followed a group of mothers and preterm infants who received skin-to-skin contact (Kangaroo Care, KC) in the neonatal period, as compared to a matched group who received standard incubator care, from birth to adulthood. Our aim was to investigate how a touch-based neonatal intervention shaped three outcomes in young adulthood: anxiety and depressive symptoms, levels of oxytocin, and markers of the immune system. Findings indicated that the effects of KC on adult outcomes were indirect and stemmed from its impact on mother, child, and the dyad across infancy, childhood, and adolescence. KC reduced maternal anxiety and depression in infancy, improved infant attention and child executive functions, and increased mother–child synchrony across development. These effects were interconnected and led to improved outcomes in adulthood. Our study uniquely describes mechanisms by which a birth intervention can influence adult outcomes through step-by-step enhancement and provides valuable insights into the topic of “developmental continuity”, a key controversy in developmental research.

**Abstract:**

Premature birth disrupts the continuity of maternal–newborn bodily contact, which underpins the development of physiological and behavioral support systems. Utilizing a unique cohort of mother–preterm dyads who received skin-to-skin contact (Kangaroo Care, KC) versus controls, and following them to adulthood, we examined how a touch-based neonatal intervention impacts three adult outcomes; anxiety/depressive symptoms, oxytocin, and secretory immunoglobulin A (s-IgA), a biomarker of the immune system. Consistent with dynamic systems’ theory, we found that links from KC to adult outcomes were indirect, mediated by its effects on maternal mood, child attention and executive functions, and mother–child synchrony across development. These improvements shaped adult outcomes via three mechanisms; (a) “sensitive periods”, where the infancy improvement directly links with an outcome, for instance, infant attention linked with higher oxytocin and lower s-IgA; (b) “step-by-step continuity”, where the infancy improvement triggers iterative changes across development, gradually shaping an outcome; for instance, mother–infant synchrony was stable across development and predicted lower anxiety/depressive symptoms; and (c) “inclusive mutual-influences”, describing cross-time associations between maternal, child, and dyadic factors; for instance, from maternal mood to child executive functions and back. Findings highlight the long-term impact of a birth intervention across development and provide valuable insights on the mechanisms of “developmental continuity”, among the key topics in developmental research.

## 1. Introduction

### 1.1. Premature Birth and Its Developmental Consequences

Premature birth often leads to substantial difficulties at the transition from childhood to adulthood [1]. Years after birth, and even among those with no gross cognitive or motor impairments, prematurely born children suffer a variety of problems, including cognitive delays [2], impaired executive functions [3,4], emotion dysregulations [5,6], social maladjustment [7,8], and behavior problems [9,10]. Several meta-analyses have shown that premature birth is associated with greater prevalence of anxiety [11] and depressive symptoms [12] across childhood and adolescence and, by young adulthood, prematurely born adults suffer more psychopathology as compared to adults born at term [10,13,14]. Similar problems have been found in endocrine support systems in children, adolescents, and adults born preterm, including aberrations in the HPA-axis stress response [15,16] and immune system functionality [17,18]. Difficulties have also been noted in attachment relationships; mother–infant interactions have been described as less sensitive and synchronous [19] and by adulthood premature infants are less likely to form romantic and close social relationships [20,21,22]. These findings underscore the broad-band, long-term impact of premature birth on physiological, behavioral, and relational systems. Given that premature birth is estimated at 11% of live births [23], amounting to millions of preterm infants born each year, the question then becomes the following: are there early interventions that can tilt the trajectory toward a more favorable course?

### 1.2. Kangaroo Care (KC) and Mother–Infant Physical Contact

Premature birth disrupts not only the maturation of the infant’s brain [24,25] but also the continuity of maternal–infant physical contact. Continuous maternal–newborn contact is critical for the development of attention and regulatory systems in humans and other mammals [26,27] and for the formation of the mother–infant bond, partly through its effects on the infant’s oxytocin (OT) system [28,29,30]. Regular physical touch between an infant and caregivers has been observed across human cultural communities in the form of co-sleeping, carrying, or daily caretaking [31]. In rodents, lack of maternal caregiving and separation of the pups from their mother caused increased defensive behavior of the animals [32] as well as elevated HPA activation [33].

The Kangaroo Care (KC) intervention for premature infants provides the only possibility for a parent to maintain full bodily contact with the fragile infant without the infant losing its body heat through the thermoregulation embedded in skin-to-skin contact. The infant is placed naked between the mother’s breasts, wearing only a diaper and a cap, in an upright position so that direct skin-to-skin contact is established [34]. KC is associated with improvement of infants’ vital signs, oxygenation level, and body temperature [35], reduces pain [36], and protects against maternal postpartum depression [37,38]. The main goal guiding our birth-to-adulthood Kangaroo Care study was to examine whether and by what mechanisms the KC intervention applied in the neonatal period impacts mental health, oxytocin, and immunity in young adulthood.

### 1.3. KC Effects across Development via Three Pathways; Maternal Mood, Child Attention and Regulation, and Dyadic Synchrony

In the current study, we utilized premature birth and the Kangaroo Care (KC) intervention as a unique “natural experiment” to pinpoint mechanisms that chart the progression from a vulnerable birth condition to adult outcomes. Results from our longitudinal study on the effects of the Kangaroo Care (KC) intervention at previous stages show that KC carries a pervasive effect on development across childhood and adolescence along three pathways; the maternal pathway, which describes the effects of KC on the mother’s mood and cognitions, particularly on reducing anxiety and depressive symptoms; the infant pathway, addressing the direct impact of contact on the infant’s attention and regulatory skills; and the dyadic pathway, which taps the effects of contact on the formation of a more synchronous mother–child relationship.

Following kangaroo contact, the maternal, child, and dyadic pathways exert mutual influences on each other over time. For instance, KC reduced the mother’s anxiety, parenting stress, and depressive symptoms in infancy and childhood [39,40,41]. Given that premature birth increases maternal anxiety and depressive symptoms in infancy [42,43], which often persist across the child’s first years [44], and better maternal mood predicts more positive child social–emotional, cognitive, and mental health outcomes [45,46], the improvement in maternal mood sets the child’s trajectory on a more adaptive course. Similarly, KC improved the child’s physiological and regulatory capacities, including focused attention, cognitive control, and emotion regulation in infancy and childhood [19,40,47]. Regulatory abilities develop on top of each other, from physiological, to emotional, to attentional, to self-regulatory capacities, and early regulation, in turn, predicts better executive functions in adolescence and improved mental health and physiology in adulthood [48,49]. Since prematurity disrupts the development of infant attention and child executive functions [50] and poor executive functions in adolescence links with a less favorable adult outcome [51,52], the improvement in focused attention in infancy following KC likely shapes better outcomes in adulthood. Finally, KC led to a more reciprocal and synchronous relationship between mother and child across development [40,47,53], indicating that the improvement in both maternal mood and child attention and regulation built a more coordinated, reciprocal dyadic process [54]. Synchrony is an individually stable feature of the relationship from infancy to adolescence [55] and young adulthood and predicts a host of favorable outcomes. As synchrony is reduced after premature birth [56], the improvement in synchrony following KC may chart a more favorable context for the child’s development over time.

The independent and joint contributions of the maternal, child, and dyadic pathways following KC and their mutual influences over time were examined in relation to three outcomes in young adulthood: (a) psychological symptoms in terms of anxiety and depressive symptoms, (b) the oxytocin system, and (c) immune system functionality as measured by levels of secretory immunoglobulin A (s-IgA), a first line mucosal barrier that serves as an index for the immune system. Studies have shown that young adults born prematurely exhibit more anxiety and depressive symptoms [10,13,57] and we expected KC to improve these symptoms, whether directly or indirectly, via the reported improvement in the mother’s mood. Oxytocin (OT) has been repeatedly linked with the parent–infant bond in humans and other mammals and its functionality is impacted by lack of contact and minimal caregiving, such as low licking and grooming [58]. Research has shown that following KC, OT levels of both caregivers and infants increased and parents with higher OT exhibited more synchrony and responsiveness during interactions [59]. OT levels and OXTR polymorphism are linked with sensitive parenting, and OT administration increases maternal sensitivity to infants in depressed mothers [60], suggesting that KC may lead to improved OT system functionality, either directly or via its impact on the mother–child relationship or the child’s attention regulation. Finally, research has shown that early life stress and stressful parenting significantly impact s-IgA levels in children. Maternal postpartum distress correlated with lower s-IgA levels in infants [61], the parent’s intrusive style was associated with children’s inflammation and immune activation [62,63], and maltreatment was linked with blunted s-IgA reactivity to acute psychosocial stress in children and adolescents [64], suggesting that the decrease in parenting stress after KC may enhance immunity.

### 1.4. Three Mechanisms of Developmental Continuity

To examine the long-term effects of the KC intervention on the three adult outcomes, we focused on the developmental cascades leading from a birth intervention to young adults’ biobehavioral outcomes. Our unique cohort, birth intervention, and longitudinal and repeated follow-up of a high-risk condition allowed us to address one of the most important topics in developmental research, that of “developmental continuity” [30,54,65,66]. Our conceptual model [67] describes three mechanisms of developmental continuity that are built on dynamic systems’ principles [68]. The first mechanism considers a “sensitive period” perspective [30], which describe the sensitivity of the system to initial conditions so that minor variations in infancy can directly impact long-term outcome. This would imply that the improvements in maternal mood, child attention, or dyadic synchrony in infancy would have a direct impact on outcomes in adulthood. The second mechanism considers “step-by-step continuity” and suggests that developmental changes in mother, child, or the dyad occur via small iterative steps and, over time, lead to measurable impact on outcome. Finally, the third mechanism considers the entire field of mutual influences of the child in context and vice versa and contends that these mutual influences are the force that gradually shapes the outcome. This mechanism reflects the dynamic systems’ notions on the ongoing interchange of organism and context in shaping development [69]. Indeed, in several longitudinal cohorts spanning birth to adolescence and young adulthood, we found that in the case of various high-risk birth conditions, such as maternal postpartum depression, premature birth, or chronic contextual trauma, effects of the early life stress on the child’s brain, behavior, and mental health were typically indirect. These effects were shaped not only by the independent contribution of child and parent but also via their mutual influences on each other over time and, especially, by altering the nature of the mother–child synchronous relationship, which is a process built from the inputs of both mother and child [70].

### 1.5. The Current Study

In light of the above, the current study examined pathways from the KC birth intervention to three outcomes in adulthood, anxiety/depressive symptoms, OT, and s-IgA, via three alternative models. These models highlight continuity in the mother, the child, or their mutual influences and were tested against a null model that did not consider the effects of KC on development, only on adult outcome. We expected that the null model, which examined direct links from birth to adulthood without considering any developmental factors, would yield the weakest fit to the data, compared to the alternative models. This would suggest that continuity passes mainly through maternal, child, and dyadic factors at different developmental stages. The next two models examined a “maternal model”, assessing the impact of KC on outcomes in young adulthood through the decrease in maternal anxiety/depressive symptoms in infancy and adulthood, and a “child model”, testing the effects of KC on adult outcomes via the improvement in the child’s attention and executive functions. We expected that both models would fit the data and show an independent contribution to outcomes through maternal and child factors. Finally, we examined a full, mutual influences inclusive model that considered maternal, child, and dyadic factors examined repeatedly across development and tested both the “step-by-step” and “sensitive period” mechanisms using factors measured across development and their mutual influences on each other. We expected the inclusive model to yield the best fit to the data, supporting the dynamic systems’ notions that the effects of a birth intervention on adult outcomes passes through mother, child, and dyadic factors measured across the entire span of humans’ protracted maturity.

## 2. Materials and Methods

This long-term study followed mother–infant dyads for 20 years. A total of 80 dyads of preterm-born young adults (mean age = 18.47 years, SD = 0.90, 45 males, 35 females) and their mothers were evaluated in the current study.

The study cohort was originally recruited in the NICU between 1996 and 1999 and included 146 premature infants and their mothers. Because prospective randomization of mothers to Kangaroo Care and control groups was not feasible for ethical reasons, as maternal bodily contact with the infant is considered a fundamental human right and the intervention is not experimental or life-threatening and to avoid a selection bias of the mother willing to take part in the KC intervention, we utilized a “natural experiment” approach during a period when the KC intervention was not widely adopted, and mothers were unfamiliar with this option. Infants were assigned to receive Kangaroo Care (KC) for 14 consecutive days (see below) or be a control with matching demographic and medical conditions. Further details on the medical and sociodemographic conditions of the infants and families and the cohort recruitment, randomization, and exclusion criteria are described in our prior publications [19,39,40,47,70].

In young adulthood, a three-hour home visit was conducted. Participants completed a demographic questionnaire, the Beck Depression Inventory [71], and State-Trait Anxiety Inventory [72]. Three saliva samples were collected: the first was a baseline sample collected after 10 min of acquaintance; the second was collected after a 15 min mother–child interaction; and the third sample was taken after a computer task. Participants received a gift certificate of NIS 150 (USD ~45) for their time. The study was approved by the university’s IRB and informed consent was obtained from all participants.

The current cohort comprised 40 participants that received KC intervention (KC group) and 40 participants that received standard incubator care during hospitalization (control group). No significant differences were found between the KC and control groups in sex, age, and birth week. However, due to attrition from the initial stages of the study, in young adulthood, there was a difference in birth weight between the groups (see Table 1) and this factor was therefore controlled in the following analysis. 

Of the original sample, consisting of 146 mother–infant dyads, we were able to visit the homes of 93 participants (63.6% of the original birth cohort). General attrition was mainly related to inability to locate families or families moving to distant locations during this 20-year-long follow-up. Of the 93 participants visited, we did not have maternal variables for two participants (2.1%), due to refusal of the mothers to participate in the last home visit. For eight dyads (8.6%), saliva collection was incomplete or damaged and an additional three dyads (3.2%) were excluded from analysis due to incomplete data.

### 2.1. Neonatal Period: Cohort and KC Intervention

The original premature cohort consisted of 146 mothers and their premature infants who were recruited in the NICU at birth between March 1996 and November 1999. This cohort was unique because it was established prior to the widespread availability of information on the Kangaroo Care (KC) intervention and its long-term benefits and, therefore, randomization was possible (see details in previous publications). The infants had an average birth weight of M = 1270 g (SD = 343.49, Range = 530–1720 g) and gestational age of M = 30.65 weeks (SD = 2.76, Range = 25–34 weeks). A total of 73 mother–infant pairs received KC and 73 were matched controls, which were carefully selected based on demographic and medical factors including gender, birth weight, gestational age, medical risk, maternal and paternal age and education, maternal employment, and parity.

Kangaroo Care Intervention: We recruited only infants who required full incubator care and could not receive full maternal–infant bodily contact outside the kangaroo position due to concern of the loss of body heat. Thus, the intervention was targeted to a period when no full bodily contact was possible between mother and infant among controls. For the KC intervention, infants were taken out of incubators, undressed, and placed between the mother’s breasts. Infants remained attached to a cardio-respiratory monitor and were observed by nurses who recorded the exact time of bodily contact. Mothers sat in a standard rocking chair and used a bedside screen for privacy. To be a part of the study, mothers committed to providing KC for at least 1 h per day for 14 consecutive days.

Nurses played a crucial role in facilitating the implementation of KC by providing instructions to parents on newborn handling and actively assisting during KC sessions. The dedicated nursing staff was committed to the well-being of the infants and families, ensuring a safe and supportive environment for KC. However, it is important to note that, at the time of the study, the KC intervention had recently been introduced to the Neonatal Intensive Care Units (NICUs) of the participating hospitals. As a result, the nurses had limited prior experience or knowledge regarding the long-term benefits associated with KC.

### 2.2. Dyadic Synchrony

Mother–child age-appropriate interactions were videotaped and scored offline, using the coding interactive behavior (CIB) system [73] to evaluate the dyadic synchronous behavior. The CIB is a well-validated system for coding social interactions that utilizes global rating scales. The CIB system has good psychometric properties and has shown construct and predictive validity, test–retest reliability, and sensitivity to cultural contexts, interacting partner, and a variety of high-risk conditions related to psychiatric conditions, child biological conditions, and environmental stress conditions. Consistent with prior research, we used the dyadic synchrony CIB construct [70,74], which is computed by averaging the CIB codes: Dyadic Reciprocity, Adaptation-Regulation, Fluency-Rhythmicity, Opportunity for Joint Expansion and Elaboration of Interaction, and Joint Positive and Relaxed Mood. These global codes across development address the dyadic feature of the interaction and express behavior in age-appropriate ways. Coding was conducted by trained coders, blind to all other information.

Infancy: Families were visited at home when infants were 3 months corrected age (mean age = 4.26 months, SD = 1.14). Mother–infant free play was videotaped in the home environment for five minutes. Instructions were “play with your infant as you normally do”. 

Childhood: When children were between 10 and 12 years, (mean age = 12.07 years, SD = 1.62) mother and child engaged in two discussion paradigms for seven minutes each, consistent with prior research [40]. In the first, a positive valance task, mother and child were asked to plan the “best day ever” to spend together, and in the second, a negative valance task, mother and child discussed a typical conflict in their relationship. 

Adulthood: In young adulthood (mean age = 18.47 years, SD = 0.91), young adults and their mothers engaged in the same positive valance task used in middle childhood and in an additional seven-minute support-giving task, in which each partner shared a topic of concern and the partner provided support. 

In infancy, synchrony is assessed by reciprocal non-verbal signals, adaptation to each other’s arousal state and social signals, rhythmic and fluent interactions, expansion of the child’s social attention and repertoire, give-and-receive reciprocity, relaxed interactions, and joint positive affect and engagement (alpha = 0.91). In childhood and adulthood, the same five codes are applied to verbal dialogue, assessing reciprocal discussion, fluent and rhythmic interactions, joint expansion of ideas and expression of feelings, relaxed and positive atmosphere, and adjustment of verbal dialogue to non-verbal signals, arousal, and communication. For adults, additional criteria include mutual and shared interaction, equal opportunity for expression, perspective taking, respect, acknowledgement, and smooth transition between adult–adult talk and non-verbal dyadic sharing (alpha = 0.85 for childhood and 0.82 for adulthood).

### 2.3. Focused Attention at 24 Months

Focused attention was assessed during cognitive testing at 24 months (corrected age) on the Bayley Scale for Infant Development, 2nd edition [75]. Item-by-item coding of the videotaped session was conducted by coders who were blind to the test results and coded each item on a scale from one to five for on-task focused attention, level of interest, orientation to object and person, impulsivity, frustration tolerance, negative affect, and positive affect. A principal components factor analysis identified two factors with an Eigenvalue above two and similar factors emerged at 12 and 24 months. The first factor loaded positively on focused attention (0.88), maintaining interest (0.84), and orientation to object and person (0.81) and negatively on negative affect (−0.73), had an Eigenvalue of 3.19, and explained 42.2% of the variance. The average of items with positive loading was used to index attention regulation capacities and was termed Focused Attention. Focused Attention had low associations with the Bayley (1993) MDI scores, r = 0.22, *p* < 0.05, and correlations emerged between delayed response and MDI scores at 24 months, r = 0.19, *p* < 0.05.

### 2.4. Executive Functions Score

At childhood, the children were tested with the Developmental Neuro Psychological Assessment (NEPSY) [76]. NEPSY is a standardized test for neuropsychological abilities in children and the executive function score was used. Executive functions address three inter-related skills: (a) inhibitory control—the capacity to resist habits, distractions, or temptations, (b) working memory—the ability to hold in mind and use information, and (c) cognitive flexibility—adjustment to change or the updating of behavior in response to changing circumstances [77]. Trained psychologists blind to the child’s group membership conducted the tests.

### 2.5. Saliva Samples Collection and Measuring 

The saliva samples were collected by passively drooling into a test tube to collect 1 to 2 mL from each participant. Overall, three samples were collected during the visit: (1) baseline (after ten minutes of acquaintance (T1)), (2) after mother–child interaction (T2; 10 min from the interaction, ~40 min from baseline), and (3) at the end of the home visit (T3; ~25 min from T2). Participants were asked not to smoke or eat for 1 h and did not drink for 30 min prior to the saliva collection. 

Saliva samples underwent three freeze–thaw cycles, (−80 °C and 4 °C), followed by centrifugation at 4000× *g* for 30 min, in order to precipitate the mucus. The supernatant was collected and stored at −20 °C.

Oxytocin: For OT analysis, the supernatant was dried by lyophilization, under vacuum at −600 °C for three days, into a cotton-like structure and stored at −20 °C until assayed. OT concentration was measured using a commercial OT Enzyme Immuno-Assay (EIA) kit (Cayman, MI, USA). On the assaying day, the dry samples were reconstructed in assay buffer of one-fourth (1/4) of the original volume, for concentered samples, which fall in the calibration curve. Measurement was conducted according to the kit’s instructions. In-house high medium and low controls were added to each plate to verify the kit’s sensitivity range. The intra-assay and inter-assay coefficients of samples and controls were less than 8.05% and 15.10%, respectively.

Secretory Immunoglobulin A (s-IgA): s-IgA concentration was measured using a commercial s-IgA- EIA kit (EUROIMMUN AG, Luebeck, Germany). Measurement was conducted according to the kit’s instructions. The inter-assay coefficients of controls and samples were less than 2.86% and 13.50%, respectively.

Visual examination of the distributions using density and q-q plots showed that OT and s-IgA were not normally distributed and these variables were log-transformed. Then, for each hormone, the area under the curve was calculated from the three measurements [78] to assess the overall hormonal production in total.

### 2.6. Anxiety/Depressive Symptoms

During the infancy and adulthood home visit, mothers filled in the Beck depression inventory (BDI) [79] and the State-Trait Anxiety Inventory (STAI) [72]. The children filled in the same questionnaires during the adulthood home visit.

BDI is a 21-item self-report inventory and it is designed to measure the severity of depressive symptoms. The STAI assesses both state and trait anxiety to evaluate anxiety while distinguishing between an anxious state at the time of assessment and an anxious personality trait. The questionnaire is composed of 40 items, and higher scores indicate greater anxiety.

To calculate the anxiety/depressive symptoms score, for each participant, the BDI and STAI trait scores were calculated and z-transformed; then, an average of the two was quantified. 

### 2.7. Statistical Analysis

Statistical analysis for group differences between KC and control was conducted using JASP software [80]. Pearson correlations assessed relationships between study variables.

### 2.8. Structural Equation Modeling

Multiple Imputation by Chained Equations (MICE) with the predictive mean matching (PMM) method was used to impute the missing data [81]. PMM has been proven to be an effective method for imputing missing data when the data are missing at random [82], which is the case in our study. 

Then, all behavioral variables were standardized and mean-centered using Z-transformation.

To examine direct and indirect associations between the neonatal intervention, maternal mental health, dyadic interactions, the preterm child’s executive functions, and the preterm adult’s mental health and biomarker outcomes, several preregistered alternative structural equation models were examined. Structural equation modeling (SEM) was performed using R programming language with the lavaan package [83] for statistical analysis. Due to a group difference in birth weight, the infant’s birth weight was used as a covariate. To assess model fit, the following indices were used: χ^2^, comparative fit index (CFI), Bollen’s incremental fit index (IFI), and the root mean square error of approximation (RMSEA). CFI and IFI ≥ 0.90 and RMSEA ≤ 0.08 values were considered to indicate a good fit [84]. Ideally, the χ^2^ statistic is expected to be non-significant in the case of adequate fit; however, this index is no longer used to evaluate fit because of its hypersensitivity to sample size [84]

### 2.9. Preregistration

The data analysis plan was preregistered following the data collection and prior to data analysis (https://osf.io/x4w3g/?view_only=d6d26973c8294bd1a0a0bf3a7b6f160d (accessed on 1 September 2022)). Deviations from preregistration include two participants (of the control group) that were excluded from the analysis due to incomplete data (more than four main variables missing). Additionally, because we had more than 25% of missing values in the mothers’ biomarkers (s-IgA and OT), these were not included in the models.

## 3. Results

### 3.1. Group Differences in Study Variables

Differences between the KC and control groups in demographic and all study variables are presented in Table 1. Independent *t*-tests revealed that KC dyads had significantly higher levels of behavioral synchrony in adulthood, childhood, and infancy. Children from the KC group had higher focused attention in infancy and a better executive function score in early adolescence. The KC mothers reported lower anxiety and depression scores in infancy (Table 2 and Figure 1), but no group differences emerged in young adulthood. Variables measured in young adulthood, anxiety/depressive symptoms, OT, and s-IgA showed no group differences. Similar analysis on the raw data (before imputation of missing variables) led to similar results with the mother–child synchrony score in infancy and childhood being marginally significant (see Appendix A). No group differences were found in demographic variables (Table 1), except in birth weight (see methods for details). To fully account for demographic differences between groups, we conducted three separate regression analyses for each of our outcome variables. No significant association was found; see Appendix A.

### 3.2. Correlations among Study Variables

Pearson correlations were used to examine associations among variables. Due to birth weight differences among groups, infants’ birth weight was entered as a covariate as seen in Table 3. Adolescents’ executive function in childhood significantly correlated with synchronous caregiving across all study time points. Maternal anxiety/depressive symptoms and dyadic behavioral synchrony were associated in infancy and childhood. In young adulthood, anxiety/depressive symptoms highly correlated with maternal anxiety/depressive symptoms and behavioral synchrony from previous and current time points. Young adults’ OT levels positively correlated with dyadic synchrony in infancy and young adulthood.

### 3.3. Path Models Linking Kangaroo Care with Adult Outcome via Developmental Cascades

Path analyses were used to test three alternative models and a null model linking the KC intervention and adults’ outcomes: anxiety/depressive symptoms, OT, and s-IgA levels.

(a) Null model. Our null model examined the direct link from KC birth to adulthood without consideration of any developmental variables. A MANCOVA test with the KC group as a fixed variable, anxiety/depressive symptoms score, OT, and S-IgA as dependent variables, and birth weight as a covariate revealed a non-significant group effect (Wilk’s lambda = 0.964, F_(3)_ = 0.73 *p* = 0.538). This indicates that the null model does not provide a good fit to the data (Figure 2A).

(b) Models via KC effects on maternal or child’s developmental variables. These models consider the links from KC to adult outcomes as mediated by either the effects of KC on the mother or its impact on the child. The maternal model (Figure 2B) examined the effects of KC on adult outcomes through its effect on reducing mothers’ anxiety and depressive symptoms. In this model, the provision of KC reduced mothers’ anxiety/depressive symptoms in infancy and this was linked with less maternal anxiety/depressive symptoms in adulthood. This improvement provided a “buffer” that predicted the three main outcomes of the child in adulthood. The fit indices for the maternal model alternative indicate a slightly better fit compared to the null model. However, the model still does not fit the data well according to the significant Chi square (χ^2^(7) = 15.191, *p* = 0.034) and RMSEA of 0.122 with lower 90% CI = 0.032 and higher 90% CI = 0.206. The Comparative Fit Index (CFI) and Tucker–Lewis Index (TLI) were below acceptable levels at 0.776 and 0.359, indicating poor model fit.

The *Child Model* (Figure 2C) assumed that KC effect is mediated by the child’s focused attention and EF. The comparative fit indices, CFI and TLI, were below acceptable levels at 0.615 and −0.284, respectively, indicating a poor fit between the model and the data. The Chi square test was significant (χ^2^(6) = 17.816, *p* = 0.007) and the RMSEA was high (RMSEA = 0.158, and lower 90% CI = 0.076 and higher 90% CI = 0.245), suggesting a poor fit of the model to the data. The regression coefficients indicate that focused attention is positively related to KC intervention and to birth weight.

Finally, the *Inclusive model* (Figure 2D) integrated mother, child, and dyadic factors for a full dynamic systems model. It includes mothers’ anxiety/depressive symptom score and multiple assessments of dyadic synchrony, as well as the child’s focused attention in infancy and executive function in childhood as mediators, to predict the three primary outcomes in young adulthood. This model showed good fit indices, with a non-significant χ^2^(27) = 33.96, *p* = 0.167, and RMSEA of 0.057 and CI = 0.000 and higher 90% CI = 0.100. The CFI and TLI were also within acceptable levels at 0.965 and 0.916, respectively.

In this model, two parallel paths leading from KC to adult outcomes and two additional converging paths were identified: The first path linked KC with fewer maternal anxiety/depressive symptoms in infancy and with fewer maternal anxiety/depressive symptoms in adulthood, which was highly correlated with the child’s fewer maternal anxiety/depressive symptoms in adulthood. A converging path from maternal anxiety/depressive symptoms in infancy was also directly associated with a lower executive functions score in childhood, which in turn was negatively associated with anxiety/depressive symptoms in adulthood.

The second path linked KC with higher mother–child synchrony in infancy, associated with synchrony in childhood and in adulthood, culminating in lower levels of child’s anxiety/depressive symptoms.

Dyadic synchrony in infancy predicted adolescents’ executive functions, which in turn linked with maternal anxiety/depressive symptoms in adulthood, which linked with the child anxiety/depressive symptoms in adulthood.

A third path associated KC with focused attention in infancy, which was negatively associated with s-IgA levels and positively associated with OT levels in adulthood.

Overall, the inclusive model showed a good fit to the data, compared with the null model and the two alternative models. Based on coefficients and model fit parameters, it can be concluded that the “inclusive model” offers the most suitable fit to the data and that the effect of KC on adulthood outcomes is indirect.

## 4. Discussion

### 4.1. Effects of a Birth Intervention on Adult Outcome via Three Mechanisms of Continuity

Results of our study are the first to chart developmental continuities from birth to adulthood across repeatedly measured physiological, attentional, regulatory, relational, and mental health factors to address the pathways by which a neonatal touch-based intervention impacts adult biobehavioral outcomes via cascading effects. We followed preterm infants and their mothers for two decades to define maternal, child, and dyadic pathways and examined how they express across development to shape outcomes. We measured three alternative models that examined how variables measured over time impact outcomes, versus a null model that examined direct effects, on three adult outcomes: anxiety and depressive symptoms, functionality of the oxytocin system, and biomarkers of immunity. Overall, our findings demonstrate that the effects of a birth intervention on adult outcomes are indirect and operate through the impact of early bodily contact on maternal, child, and dyadic factors across childhood. Three mechanisms of developmental continuity were found to operate in shaping adult outcome: the “sensitive period” mechanism, the “step-by-step” mechanism, and the full butterfly “mutual influences” mechanism that considers the influences of the child in context and vice versa.

We found that developmental cascades can ride on multiple pathways to link early-life experiences with long-term biobehavioral outcomes. Results indicated that KC altered the three pathways—the maternal, child, and dyadic pathways—already in infancy; it improved maternal anxiety and depressive symptoms, augmented the infant’s focused attention, and boosted mother–infant dyadic synchrony and the three separate effects influenced each other over time. Indeed, our findings show that the inclusive butterfly model of mutual influences provided the best fit to the data. This model integrated the three pathways impacted by KC and showed how each pathway triggered long-term effects through the “sensitive period” or the “step-by-step” mechanisms of developmental continuity.

### 4.2. KC Impact on Adult Outcome via “Maternal Effects”

The first pathway from KC to outcome operated via “maternal effects”, by reducing maternal anxiety and depression in the first months of parenting. Following premature birth and the ensuing incubation, mothers report increased strain, hopelessness, disappointment, guilt, and emotional instability [85], which often increase with the separation and potential loss of the child. Importantly, while in the general population postpartum depression prevalence is approximately 15% [86], following the birth of a very low-birthweight preterm infant, 20% of mothers reported clinically significant depression while 43% had moderate to severe anxiety [42] and a review of the literature indicates that premature birth places mothers at a greater risk to develop postpartum depression [87].

We found that KC reduced the mother’s anxiety and depression in the first postpartum period, which, in turn, linked with lower maternal anxiety and depressive symptoms in adulthood and was associated with lower anxiety and depression in their adult offspring. In addition to this significant mediation in the maternal pathway, we also found a “crisscross” effect of mother on child and vice versa. The mother’s improved mood in infancy linked with better child executive functions at the transition to adolescence, which, in turn, impacted maternal mood in adulthood, culminating in lower anxiety and depression in young adults born preterm, indicating how maternal anxiety/depressive symptoms are interwoven with the child’s EF in a mutually influencing, cross-time pattern. Such mutual influences from the mother’s early anxiety and depression to the child’s EF are consistent with studies showing that maternal sensitivity, autonomy-granting behavior, and mind-mindedness predicts children’s EF 6–12 months later [88], and different aspects of parent–child interactions affect different EF domains in later childhood [89]. Other studies showed that maternal pre- and postnatal depression and anxiety symptoms longitudinally impact poor child attention, lower academic achievements, and decreased EF abilities [90,91,92]. Our findings add the complementary impact of the child’s EF skills on maternal anxiety and depressive symptoms in young adulthood, suggesting that caring for a more regulated child carries mental health benefits for the mother.

### 4.3. KC Impact on Adult Outcome via “Child Effects”

A second pathway from the KC intervention to outcome involved “child effects” that included improvements in the child’s focused attention in infancy and executive functions at the transition to adolescence, both of which were directly and indirectly related to the three outcomes. We found that better focused attention at 24 months—the child’s ability to maintain on-task persistent attention, focus on object and experimenter, maintain neutral affect during the execution of a cognitive task, and attend to all stages of the problem at hand—was directly related to higher OT and lower s-IgA levels in young adulthood. Our results are consistent with previous studies which point to associations between child attention or EF and long-term outcome. For instance, low executive functions predicted school-aged children’s behavioral and emotional problems and difficulties with peers as reported by both parents and teachers [93] and prematurely born preschoolers showed lower EF compared to children born at term even when controlling for IQ scores [94]. Furthermore, the effects of adverse early experiences and psychopathology on health-risk behavior was found to be mediated by executive functions [95] and EF difficulties are thought to contribute to the high prevalence of psychopathology in the preterm population [11]. Our findings highlight the measurable, broad-band impact of KC on outcomes via its effects on the child’s early attention regulation and EF.

### 4.4. KC Impacts on Adult Outcome Via “Dyadic Effects”

Finally, the KC intervention increased behavioral synchrony between mother and child and these differences remained significant across adolescence and up to young adulthood, despite the fact that the early non-verbal synchrony becomes more complex, verbal, empathic, and mutual as infants grow and assume greater responsibility for the interaction [96]. In a previous study [70], we found that in young adulthood, mother–child synchrony in the KC group was similar to that of a full-term comparison group, suggesting that the improvement in the mother–infant relationship following KC carries a long-term effect throughout development. Notably, synchrony operated via both the “step-by-step” and “sensitive period” mechanisms. Synchrony in infancy linked with synchrony in adolescence and adulthood and higher synchrony in adulthood, in turn, was associated with lower anxiety and depression in young adults, consistent with the step-by-step iterative mechanism. In addition, synchrony in infancy directly linked with greater child OT levels in adulthood, suggesting that, similar to findings in animal models, which showed that better parenting (i.e., higher licking and grooming) permanently altered the offspring’s OT system. Prior research has similarly shown that the KC intervention is associated with increases in maternal and paternal OT and a decrease in cortisol [97]. These findings highlight the “sensitive period” effect from synchrony experienced at 3–9 months, the sensitive period for maturation of the social brain [98], to oxytocin system functionality in adulthood. Such findings are consistent with those described in other mammals but rarely found in humans due to the very lengthy maturation of the human infant and the difficulty in following infants across two decades. Mother–preterm interactions have been described as less optimal, characterized by higher intrusiveness, more negative affect, and lower reciprocity [99]. Hence, the long-term effects of the improvement in mother–infant synchrony on adult biology and psychopathology highlight the importance of enhancing the mother–infant relational quality following premature birth.

### 4.5. KC, Oxytocin, and S-IgA

The effects of KC on oxytocin and s-IgA via improving infant attention and the mother–child synchronous relationship are consistent with studies in animal models. Maternal separation is an unnatural situation for mammalian young. In lab animals, unpredicted maternal separation is often used as an early-life stressor. By removing maternal presence and its stress-suppression qualities from the pups, separation serves as a potent naturalistic stressor that carries long-term behavioral and physiological effects. In rodents, six hours of daily maternal separation between 2 and 20 postnatal days resulted in an increase in reactive corticotropin releasing factor (CRF) concentration and CRF receptor density, suggesting a long-term effect on the stress response [100]. Maternal separation also alters typical maternal behaviors, such as licking and grooming, and reduces the number of OT and CRF immunoreactive cells in the paraventricular nucleus [58], highlighting the effects of maternal separation on both the offspring OT and immune system. Mothers of preterm infants have higher s-IgA levels in breastmilk and feelings of perceived stress, anger, and depression were associated with s-IgA levels in postpartum mothers [101,102,103]. Our findings on the long-term indirect impact of KC on OT and s-IgA are therefore meaningful and require further study.

In addition to maternal separation, studies have underscored the associations between s-IgA and anxiety symptoms in children [104]. In our lab, we found that exposure to a variety of high-risk birth conditions increases children’s s-IgA, including children growing up in the context of chronic maternal depression and those living in zones of chronic war-related trauma [105,106]. Furthermore, we found that s-IgA levels are associated with dysregulated temperament in young children [63]. Higher s-IgA levels were found in children with ADHD [107] and meditation-based training led to a shorter reaction time in executive attention tasks and to higher s-IgA levels following stress [108]. These findings are consistent with our results on the associations between improved attention regulation in infancy and better functionality of the immune system in adulthood.

While both the “maternal” and “child” models showed some significant paths, the inclusive model, which considered the effects of all three pathways, the maternal, child, and dyadic pathways, and measured the entire field of their mutual influences as they emerge across development, provided the best fit for the data. This highlights the multi-domain, long-term effects of a touch-based intervention during the immediate post-birth period on mother, child, and the dyad during the “sensitive period” of massive plasticity and show how these immediate effects trigger iterative processes by which mother and child influence each other and gradually shape outcomes. Our findings underscore the importance of early life interventions for preterm infants and their caregivers to maximize their effects on developmental outcomes.

## 5. Limitations

While this study provides valuable insights into the effects of KC on the development of premature infants, several limitations should be acknowledged. Firstly, our exclusion criteria, including factors such as intraventricular hemorrhage (IVH), may have influenced the composition of the study sample. By excluding infants with severe neurological damage, such as IVH, who often present additional medical complications, the generalizability of our findings to this specific population may be limited. Future research should aim to include a broader range of infants, including those with complex medical conditions, to examine the potential benefits of KC in these cases.

Furthermore, the demographic conditions of the participating families may affect the generalizability of our results to other populations. Our study included families with specific characteristics and excluded conditions such as teenage pregnancy or single parenting. These factors may introduce bias and limit the extrapolation of our findings to different socio-demographic groups. Future studies should strive to include a more diverse sample to enhance the generalizability of the results.

Additionally, it is important to acknowledge that the study was conducted within a specific timeframe and in particular healthcare settings. The characteristics and resources available in these settings, such as parents’ encouragement for active participation in infant care routines, may differ from those available in other healthcare facilities. Therefore, caution should be exercised when generalizing the findings to settings with different levels of care or limited parental involvement.

Moreover, there are specific limitations to consider. Firstly, there was a birthweight difference between the groups due to natural attrition over the 20-year span. However, this difference was controlled in the analysis. Secondly, the sample size was relatively small, which may have limited the power to detect group differences in some outcome variables. Lastly, the absence of executive function (EF) measures from adulthood prevents a more accurate and recent evaluation of EF in the adult participants.

Despite these limitations, our study is the first long-term follow-up of a birth intervention and provides unique insights into developmental trajectories leading to mental health, immune, and hormonal outcomes. Much further research is needed using larger samples that are followed over time that include multiple domains of assessment to fully understand the complex process of developmental continuity across humans’ protracted maturity from birth to adulthood.

## 6. Conclusions

Utilizing a unique birth cohort of mother–preterm dyads followed from birth to young adulthood, we examined the long-term effects of a neonatal touch-based intervention. We found that the effects of the intervention on biobehavioral outcomes in adulthood are indirect and consistent with dynamic systems’ models on “developmental continuities” [109]. These included the effects of the intervention on mother, child, and the dyad, which conformed to both the “step-by-step” iterative continuity and the “sensitive period” mechanisms. Much further research is needed to follow humans across their protracted maturity and assess the multiple ways in which mother and child impact each other over time toward the construction of targeted interventions.

## Figures and Tables

**Figure 1 biology-12-00847-f001:**
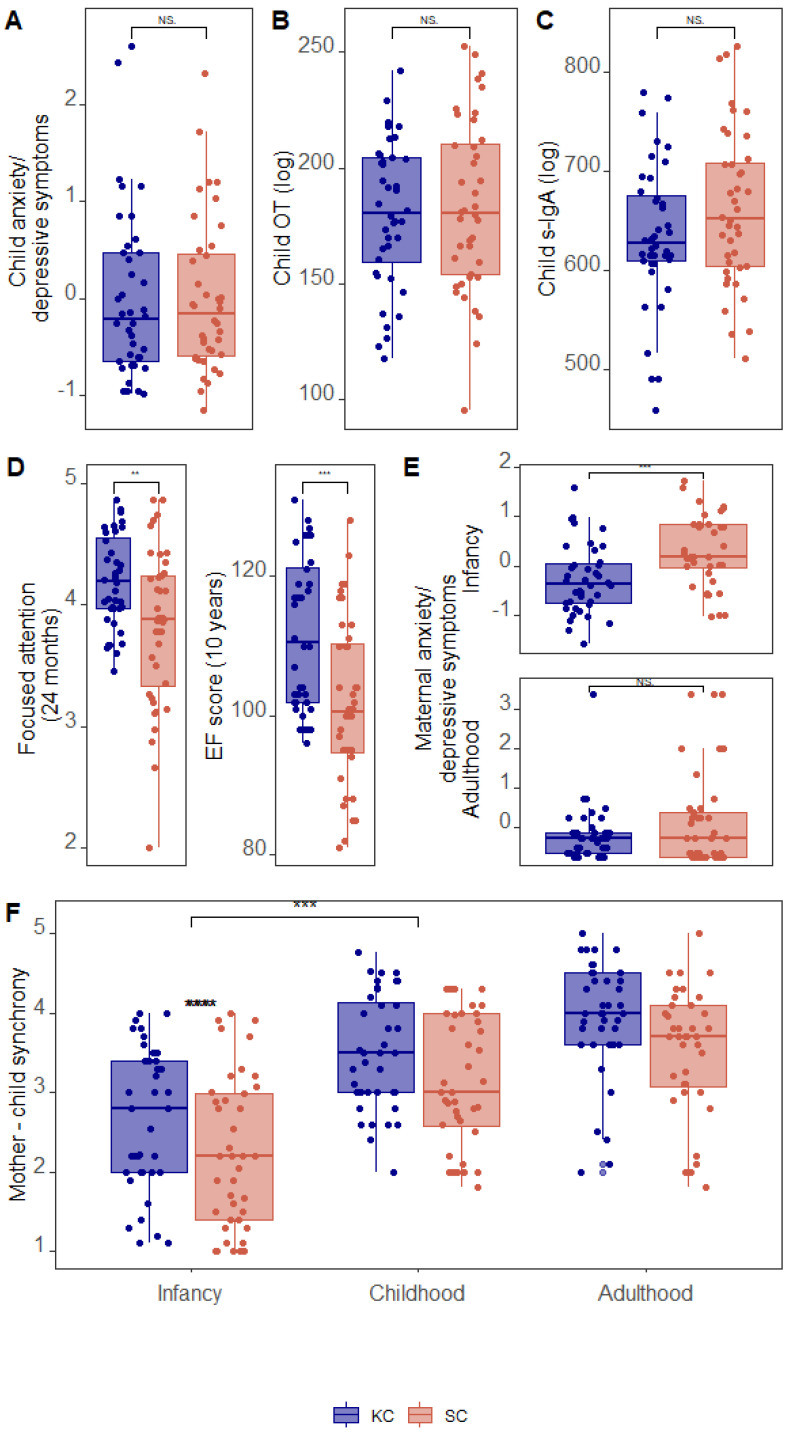
Differences between groups (KC = Kangaroo Care; SC = standard care, control group) in study variables. (**A**–**C**): child outcomes in adulthood: (**A**) Child anxiety/depressive symptoms. (**B**) Child OT level. (**C**) Child S-IgA levels. (**D**) Child focused attention at 24 months (**left**) and NEPSY executive functions score at 10 years (**right**). (**E**) Maternal anxiety/depressive symptoms in infancy (**top**) and adulthood (**bottom**). (**F**) Mother-child dyadic synchrony from infancy to adulthood. T-tests significance levels: ** *p* < 0.01, *** *p* < 0.005, **** *p* < 0.0001. NS: not significant.

**Figure 2 biology-12-00847-f002:**
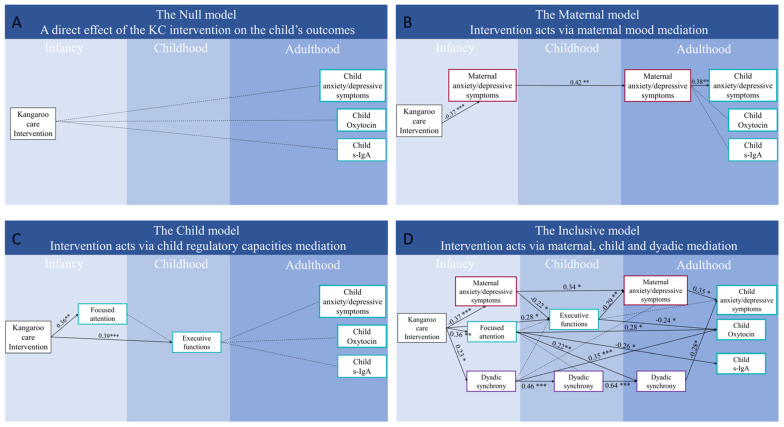
Proposed models for the association of KC with adult outcomes. Path analysis linking KC group with young adults’ mental health/emotional distress, OT, and S-IgA levels with (**A**) no mediators (the null model), (**B**) maternal emotional distress as mediator, (**C**) child’s focused attention and executive functions as mediators, and (**D**) maternal emotional distress, mother–child dyadic synchrony, child’s focused attention, and EF as mediators. Coefficients represent standardized regression weights. * *p* < 0.05, ** *p* < 0.01, *** *p* < 0.005. All models are controlled for infant birth weight.

**Table 1 biology-12-00847-t001:** Comparisons between groups in main demographic variables.

	KC	Control	t/χ^2^	Cohen’s d
**Gender (% Female)**	19 (47.5%)	16 (40%)	χ^2^_(1)_ = 0.457, *p* = 0.499	
**Child age**	18.49 (SD = 0.10)	18.45 (SD = 0.84)	t_(78)_ = 0.218, *p* = 0.82	0.049
**Mother age**	48.2 (SD = 4.46)	48.5 (SD = 5.12)	t_(67)_ = −0.428, *p* = 0.67	0.103
**Mother education (above average)**	22 (62.9%)	26 (70.3%)	χ^2^_(4)_ = 0.178, *p* = 0.776	
**Household income (above average)**	12 (30.76%)	12 (33.33%)	χ^2^_(2)_ = 4.135, *p* = 0.13	
**Birth week (weeks)**	30.769 (SD = 2.76)	31.475 (SD = 2.43)	t_(78)_ = −1.254, *p* = 0.21	−0.280
**Birth weight (gram)**	1296 (SD = 278)	1460 (SD = 356)	**t_(78)_ = −2.301, *p* = 0.024**	**−0.514**

Bold marks a significant group difference.

**Table 2 biology-12-00847-t002:** Comparisons between groups in main study variables.

	KC Mean (SD)	Control Mean (SD)	*t*-Test	Cohen’s d
**Child depression/anxiety**	−0.04 (0.86)	−0.01 (0.73)	t_(78)_ = −0.155, *p* = 0.87	−0.009
**Child OT**	180.13 (31.2)	183.52 (36.9)	t_(78)_ = −0.442, *p* = 0.66	−0.099
**Child s-IgA**	634.79 (72.4)	661.91 (79.0)	t_(78)_ = −1.599, *p* = 0.11	−0.358
**Child Focused attention**	4.19 (0.39)	3.82 (0.64)	t_(78)_ = 3.157, *p* = 0.002	0.706
**Child EF score**	111.87 (11.06)	101.85 (11.9)	t_(78)_ = 3.902, *p* < 0.001	0.873
**Mother depression/anxiety**	−0.27 (0.51)	0.30 (1.08)	t_(78)_ = −1.43, *p* = 0.157	−0.320
**Mother depression/anxiety (infancy)**	−0.19 (0.71)	0.08 (0.71)	t_(78)_ = −3.580, *p* < 0.001	−0.800
**Dyadic synchrony infancy**	2.66 (0.87)	2.23 (0.95)	t_(78)_ = 2.108, *p* = 0.038	0.471
**Dyadic synchrony childhood**	3.49 (0.71)	3.13 (0.82)	t_(78)_ = 2.073, *p* = 0.041	0.464
**Dyadic synchrony adulthood**	3.95 (0.74)	3.51 (0.80)	t_(78)_ = 2.537, *p* = 0.013	0.567

**Table 3 biology-12-00847-t003:** Pearson correlations between child, maternal, and dyadic variables.

	Dyadic Synchrony			Child Regulatory Capacities		Maternal Anxiety/Depression		Child—Adulthood Outcomes		
	Infancy	Childhood	Adulthood	Focused Attention	EF Score	Infancy	Adulthood	Emotional Distress	s-IgA	OT
**Dyadic synchrony** (Infancy)	—									
**Dyadic synchrony** (Childhood)	0.506 ***	—								
**Dyadic synchrony** (Adulthood)	0.490 ***	0.762 ***	—							
**Focused attention**	0.215	0.300 **	0.431 ***							
**Child EF score**	0.341 **	0.225 *	0.161							
**Maternal depression/anxiety** (Infancy)	−0.298 **	−0.264 *	−0.261 *	−0.172	−0.386 ***	—				
**Maternal depression/anxiety** (Adulthood)	−0.095	−0.172	−0.073	−0.074	−0.387 ***	0.451 ***	—			
**Child depression/anxiety**	−0.182	−0.260 *	−0.248 *	−0.169	−0.231 **	0.420 ***	0.376 ***	—		
**Child s-IgA**	−0.115	0.143	0.004	−0.259 *	−0.084	0.126	−0.059	−0.016	—	
**Child OT**	0.321 **	0.179	0.299 **	0.289 **	−0.051	0.052	−0.120	−0.019	0.03	—

* *p* < 0.05, ** *p* < 0.01, *** *p* < 0.001.

## Data Availability

The data that support the findings of this study will be shared upon request from the corresponding author.

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
