# Peer review of "Developmental Cascades Link Maternal–Newborn Skin-to-Skin Contact with Young Adults’ Psychological Symptoms, Oxytocin, and Immunity; Charting Mechanisms of Developmental Continuity from Birth to Adulthood"

_biology, 2023, doi:10.3390/biology12060847_

Round 1
Reviewer 1 Report
This study uses a multiple modality / pathway approach to identify the impact of a short duration of Kangaroo care when preterms were stable for 2 weeks for at least 1 hour a day, and followed them until 18 years.
the sample size has fallen from recruitment 74 per group to 40 per group now. Reasons for loss to FU now should be mentioned. You state that the unseen are NS different to those seen.
I've read the initial recruitment and description of the study form the 2002 paper. this is not a randomised study but a contemporary matched cohort study at 2 similar hospital sites. Please amend Line 171 and 172 to reflect this.
The following paper outlines the full study better than ref 18. (which is dated 2007, but is 2003 and only included 35 in each arm.
- Skin-to-Skin contact (Kangaroo care) promotes self-regulation in premature infants: sleep-wake cyclicity, arousal modulation, and sustained exploration. Dev Psychol. 2002 Mar;38(2):194-207. doi: 10.1037//0012-1649.38.2.194.Dev Psychol. 2002. PMID: 11881756 Did the KC nurse role during the KC sessions include education and support, that added to the physical KC. ie given it is a short intervention and small study size, the effect is persistent. A section on the study limitations should also mention the effect of exclusions included IVH, and the demographics of the families on generalisability. The following paper is a more recent IPD metaanalysis than ref 13 EClinicalMedicine 42 (2021) 101216 Peter J Anderson, PhDa,b,*, et al, Psychiatric disorders in individuals born very preterm / very low-birthweight: An individual participant data (IPD) meta-analysis. https://doi.org/10.1016/j.eclinm.2021.101216
Reviewer 2 Report
This manuscript addresses longitudinal outcomes and pathways leading from skin-skin contact between mothers and infants in the neonatal ICU. Its longitudinal nature and the number of outcomes studied makes it a complex manuscript. In its current form it is difficult to read, and requires improvements particularly to the structure of the introduction and discussion. Some aspects of the analysis strategy and description of methods require justification or clarification. My specific comments are below.
1. The introduction is disorganised. Section headings and restructuring would be helpful. It begins appropriately with why premature birth is a problem requiring intervention, but a complete summary of what the study was intended to achieve should be in the first paragraph rather than interspersed between six paragraphs.
2. Kangaroo care (KC) is not clearly defined. Nor does the introduction place it in a wider context of the importance of physical contact between mothers and infants for infant outcomes and the strength of the bonds between mothers and infants. This could include for example, studies of bedsharing/co-sleeping on infant outcomes such as stress, and moving some of the evidence from studies of non-humans which are introduced in the discussion section.
3. Related to the point above, 15/65 (23%) of studies cited in the introduction are papers written by or contributed to by one of the authors (Feldman). While I understand that the manuscript is part of a long term project with many research outputs, the introduction should be built on a broader framework of past studies.
4. There is little information in the Methods section on KC care: we don’t know how many times infant had skin-skin contact or for how long. We also don’t know what care occurred in the control group infants.
5. On lines 187-188. How was birth weight controlled for in the analysis? And in which analysis was it controlled? I assume controlling for birth weight was performed by calculating standardised residuals, but detail is needed.
6. The results should additionally include effect size estimates for the t-tests, such as Cohen’s d, so that readers can assess the magnitude of each difference. While I note that birth weight was controlled for due to the difference shown in Table 1, differences between KC and controls in baseline variables are not the only reason why you should statistically control for all of the variables shown in Table 1. For example, IgA levels and oxytocin receptor function are known to be associated with socioeconomic status, and ideally you would seek to control for these effects in looking at the association of KC care with each outcome measure. Hence, rather than t-tests, you should ideally use multiple regression to test for differences between KC and controls.
7. What is contained in the parentheses in Table 2? Is it SE?
8. The discussion needs overt structure and headings.
A few useful references related to my comments:
Phil Evans, Geoff Der, Graeme Ford, Frank Hucklebridge, Kate Hunt, Shirley Lambert, Social Class, Sex, and Age Differences in Mucosal Immunity in a Large Community Sample, Brain, Behavior, and Immunity, Volume 14, Issue 1, 2000, Pages 41-48, ISSN 0889-1591, https://doi.org/10.1006/brbi.1999.0571. Elaine S. Barry, Co-sleeping as a proximal context for infant development: The importance of physical touch. Infant Behavior and Development, Volume 57, 2019, 101385, ISSN 0163-6383, https://doi.org/10.1016/j.infbeh.2019.101385 Ann Bigelow, Michelle Power, Janis MacLellan‐Peters, Marion Alex, Claudette McDonald, Effect of Mother/Infant Skin‐to‐Skin Contact on Postpartum Depressive Symptoms and Maternal Physiological Stress. Journal of Obstetric, Gynecologic & Neonatal Nursing,Volume 41, Issue 3, 2012, Pages 369-382, ISSN 0884-2175, https://doi.org/10.1111/j.1552-6909.2012.01350.x
Round 2
Reviewer 2 Report
The authors have comprehensively addressed the issues that I raised.